# Ranking of non-coding pathogenic variants and putative essential regions of the human genome

Alex Wells [1], David Heckerman [2], Ali Torkamani [3], Li Yin [3], Jonathan Sebat [4,5,6], Bing Ren [7], Amalio Telenti[3,8,9,10]* & Julia di Iulio [3,9,10]*

A gene is considered essential if loss of function results in loss of viability, fitness or in disease. This concept is well established for coding genes; however, non-coding regions are thought less likely to be determinants of critical functions. Here we train a machine learning model using functional, mutational and structural features, including new genome essentiality metrics, 3D genome organization and enhancer reporter data to identify deleterious variants in non-coding regions. We assess the model for functional correlates by using data from tiling-deletion-based and CRISPR interference screens of activity of *cis*-regulatory elements in over 3 Mb of genome sequence. Finally, we explore two user cases that involve indels and the disruption of enhancers associated with a developmental disease. We rank variants in the non-coding genome according to their predicted deleteriousness. The model prioritizes non-coding regions associated with regulation of important genes and with cell viability, an in vitro surrogate of essentiality.

[1] Stanford University, Stanford, CA 94305, USA. [2] Department of Computer Sciences, University of California Los Angeles, Los Angeles, CA 90024, USA. [3] Scripps Research Translational Institute, La Jolla, CA 92037, USA. [4] Beyster Institute for Psychiatric Genomics, Department of Psychiatry, University of California San Diego, La Jolla, CA 92093, USA. [5] Department of Cellular and Molecular Medicine, University of California San Diego, La Jolla, CA 92093, USA. [6] Department of Pediatrics, University of California San Diego, La Jolla, CA 92093, USA. [7] Ludwig Institute for Cancer Research, La Jolla, CA 92093, USA. [8] Department of Integrative Structural and Computational Biology, The Scripps Research Institute, La Jolla, CA 92037, USA. [9] Present address: Vir Biotechnology, Inc., San Francisco, CA 94158, USA. [10] These authors contributed equally: Amalio Telenti, Julia di Iulio. *email: atelenti@scripps.edu; Julia.diiulio@gmail.com

There is rapid improvement in the understanding of the human genome, the organization of function, and the consequences of human genetic variation. This understanding enables multiple innovations in medical genetics. For example, exome sequencing accelerates the diagnosis and contributes to the clinical management of rare genetic disorders. However, exome sequencing only examines less than 2% of the genome sequence and the current diagnostic yield stands at around 30%[1–4]. The role of genetics in many other rare disorders is at present unknown, and the exact mechanisms of disease remain largely unexplored. One potential mechanism is through perturbation of important regulatory regions of the genome. Recent efforts at extending the search space from the coding to the immediate regulatory regions show that new pathogenic variants can be identified in a small fraction of cases[5–7]. In parallel, there are recent reports of diseases that implicate distal enhancers and changes in the three-dimensional (3D) genome structure[6,8]. Thus, the next milestones in the interpretation of the human genome sequence will emerge from the analysis of functional consequences of genetic variants in the non-protein coding (here in referred to as non-coding) genome—the remaining 98% of genome sequence that includes the regulatory machinery. As previously done for coding genes[9], we define a non-coding genomic element as putative essential when loss of its function may compromise viability of the individual or results in profound loss of fitness or in disease.

Interpretation of the non-coding genome requires the identification of landmarks, features and structures, the same principles that aid the interpretation of the coding genome. Genome-wide epigenomic maps have revealed hundreds of thousands of regions showing signatures of enhancers, promoters and other gene-regulatory elements[10]. However, the high-resolution dissection of functionally relevant nucleotides as well as the hallmarks of essentiality in the non-coding genome remain limited at present[11]. Multiple sources of biochemical, genetic and evolutionary data convey functional information on the non-coding genome[12]. These data are used by different scoring algorithms[13–22] that aim at ranking variants according to their predicted deleteriousness. The accuracy of these methods typically increases with ensemble-based classifiers[23] which integrate multiple models. The precision of functional and deleteriousness prediction can be further increased by learning from novel data sources. Sources of data that have not been included in previous analyses include studies of the patterns of human-specific constraints that are revealed by population genomic analyses[24], analyses of 3D organization of the genome (e.g., promoter capture Hi-C)[25,26], and from high-throughput screens of enhancer function[27]. In this work, we implement state-of-the art machine learning tools to rank-classify putative essential elements of the non-coding genome with an emphasis on the contribution of new data modalities. We then use tiling array deletion and CRISPR interference (CRISPRi) data to assess the possible functional relevance of the predictions. Lastly, we assess the predictive tool in two clinical settings: structural variants associated with the autism spectrum disorder (ASD) and in a developmental disorder resulting from the disruption of the regulatory machinery. The study provides a tool for the interpretation of variants in the non-coding genome and for the prioritization of genomic regions that are associated with critical functions. The study design is summarized in Supplementary Fig. 1.

## Results and Discussion

**Training a model to identify putative essential non-coding elements.** To train a supervised machine learning model, we included non-coding pathogenic variants from ClinVar[28] and Human Gene Mutation Database (HGMD)[29] ($N = 782$, Supplementary Data 1 and Supplementary Fig. 2). The set of control variants was built by using all variants from gnomAD (http://gnomad.broadinstitute.org/) with allele frequencies >1% across populations and sub-selecting ($N = 9516$) those that matched the pathogenic variant set based on distance to splice sites and genomic element distribution. For validation, we used non-coding pathogenic variants not included in the original dataset, from a new release of ClinVar and HGMD (total of $N = 286$, including $N = 77$ mapping to non-coding RNAs (ncRNAs); see Methods, Supplementary Data 1 and Supplementary Fig. 2). To mitigate bias in the selection of pathogenic and control variants, we use multiple strategies: (i) the control variants are selected to be the closest common variants matching the pathogenic variant in terms of genomic element and distance to splice site, (ii) a minimum distance of 500 bp is allowed between pathogenic variants to prevent overweighting of some genomic regions in the model, but still allowing for sub-genic genomic element resolution, (iii) the model is trained and tested on non-overlapping chromosomic regions to counteract the potential over-training of some genes.

We trained an XGBoost model, an implementation of gradient-boosted decision trees consisting of a collection of decision trees, where a node in a single decision tree splits the training data into subsets (deleterious versus benign). During testing, new variants with the same feature sets were given to each tree to make a prediction (putative essential or non-essential). The outputs of each tree were combined ("ensembling") to generate a final prediction. Each variant in the dataset was annotated with 38 features from four major categories (Supplementary Data 2). (i) Essentiality features, such as context-dependent tolerance score (CDTS)[24] and probability of loss-of-function intolerance (pLI)[30], among others. The latter was used by mapping each non-coding genetic variant to the closest gene and assigning the gene essentiality score of that gene to the corresponding variant. (ii) Chromatin structure features, such as chromosome conformation[24,26] data used either as a binary indicator to denote whether or not a given non-coding genomic position physically interacts with gene promoters, or as a continuous feature, by attributing the respective gene essentiality of the associated promoter to the distal interacting region. The loop and anchor features were also used as discrete values representing the number of cell lines where they were identified. (iii) Gene expression-related features, such as readout of high-throughput enhancer functional screens[27], and (iv) existing non-coding deleteriousness metrics: CADD[14], ncEigen[15], FATHMM[18], FunSeq2[17], LINSIGHT[22], ORION[21], ReMM[19] and ncRVIS[31].

We scored the non-coding regions genome-wide. The result of this process was a score (ncER, non-coding essential regulation) for each nucleotide, ranging from 0 (non-essential) to 1 (putative essential). We evaluated the model performance on a test set comprising 20% of the data through fivefold cross validation and assessed the generalization of the model on two independent non-overlapping sets, consisting of hold-out HGMD and ClinVar variants, for validation of the performance of the classifier (see Methods, Supplementary Data 1 and Supplementary Fig. 2). ncER reached a receiver operating characteristic (ROC) AUC of 88% and a precision-recall (PR) AUC of 41% on the test set (Fig. 1). We then aimed at understanding the collective and individual contribution of the new features, namely essentiality, 3D genome organization and gene expression features, compared to the prediction achieved with all previously used scores. A model trained solely with the new features performed similarly to a model trained with previously published metrics (Fig. 1a, b, Supplementary Fig. 3), but most importantly the addition of the new features in the model increased both ROC and PR AUCs by

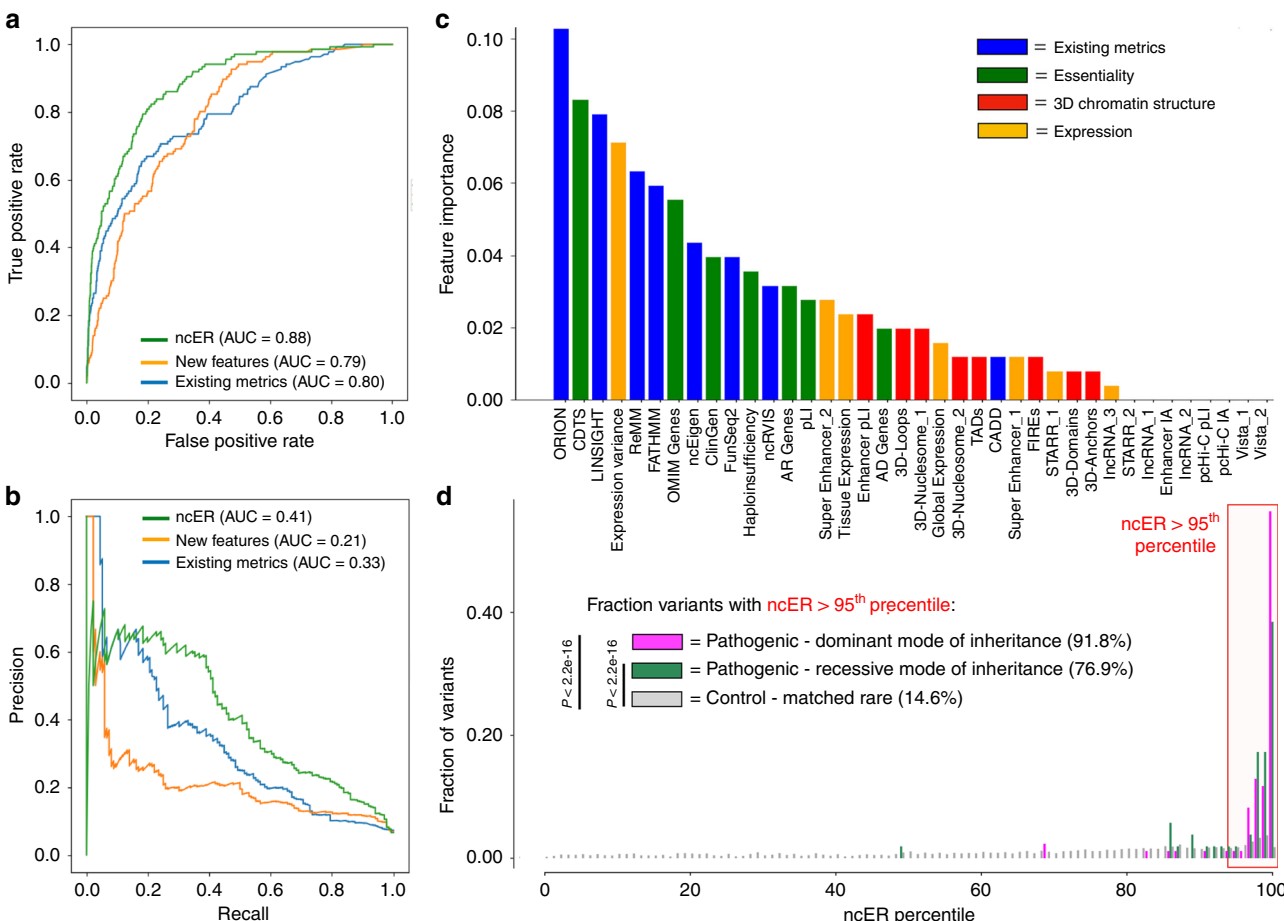

**Fig. 1** Ensemble learning for the prediction of deleterious variants in the non-coding genome. Performance ROC-AUC (**a**) and PR-AUC (**b**) on the test set ($N = 136$ non-coding pathogenic and $N = 2017$ control variants) of a model trained only with published deleteriousness metrics (blue), only with new features, namely essentiality, 3D genome organization and gene expression features (orange) and with both new features and published metrics (green, ncER). The importance of the various input features in the ncER model is shown in **c**. Blue, published scores; green, new essentiality features; red, new 3D chromatin structure features; orange, new regulatory/functional screen features. Panel **d** shows the distribution of ncER percentiles for an independent set of 137 curated non-coding pathogenic Mendelian variants compared to a set of singletons from gnomad matched by genomic element and distance to splice sites. There is statistically significant enrichment for both dominant ($N = 85$) and recessive ($N = 52$) non-coding pathogenic variants in high ncER percentiles. p Values were computed with Fisher's exact test. ROC receiver operating characteristic, PR precision-recall, AUC area under the curve

at least 8%. The univariate importance of each input feature in the model is displayed in Fig. 1c. Most of the features (30 out of 38) in the model contributed to the score. The top contributing new features were Orion[21] and CDTS[24] that measure human-specific genomic constrain. Because of sparsity of some of the data (for example, pcHi-C and Vista enhancers), some of the features did not contribute to the model as few variants mapped to informative positions. The model generalized to the independent validation datasets, including to a set of pathogenic variants mapping to ncRNAs, achieving an ROC-AUC ranging from 84 to 93% and a PR-AUC ranging from 45 to 65% (Supplementary Fig. 4).

We then examined the nature of the high ncER score regions of the genome based on different ncER percentile thresholds (99.9th, 99.5th, 99th and 95th percentiles, representing 1.7, 9.9, 21.4 and 117 Mb of cumulative sequence not overlapping protein coding regions). The distribution of genomic elements at each threshold is displayed in Supplementary Figs. 5A and 5B. All types of genomic elements were represented in the highest ncER score bins of the genome, although *cis*-regulatory and enhancer sequences were enriched in the highest percentiles. High ncER score regions were of small size, with the most common size range being single nucleotides (Supplementary

Fig. 5C). To have an overview of the putative function of high ncER score regulatory regions, we did pathway analyses for the set of genes ($N = 701$) with at least one promoter nucleotide in the top 99.9% ncER values. The most significant enriched biological process GO terms included development and regulation of gene expression, such as heart development and negative regulation of gene expression (Supplementary Fig. 6). Finally, we examined the distribution of ncER percentile for an independent curated set of non-coding Mendelian variants (see Methods, Supplementary Data 1 and 3, and Supplementary Fig. 2). The majority of mutations (86%) had ncER values above the 95th percentile, and 58% above the 99th percentile; a 5.9- and 15.4-fold enrichment over matched singleton variants, respectively (Fig. 1d). Similarly, we observed a significant shift to higher ncER values when assessing genome-wide association studies (GWAS) hit single-nucleotide variants (SNVs) (see Methods, Supplementary Data 1 and Supplementary Fig. 2) compared to a control set of common variants (Supplementary Fig. 7).

In summary, a model that trains on novel genomic features (essentiality, 3D organization, expression) adds precision to previous models that trained on partially orthogonal features (biochemistry, conservation). The model performs well in testing

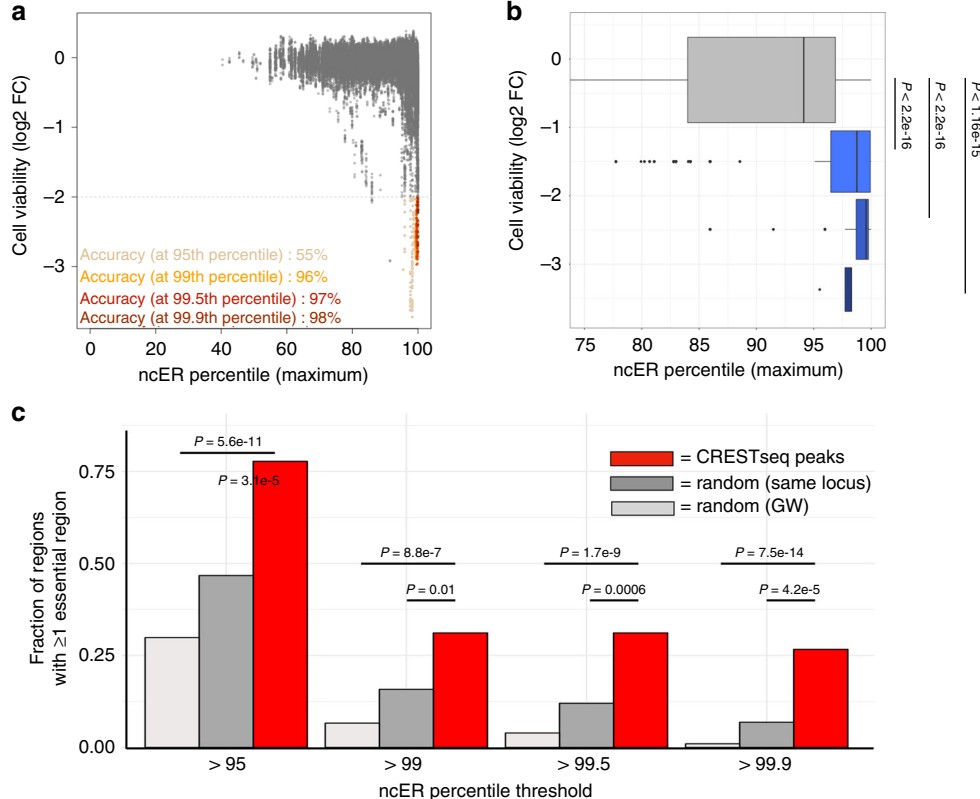

**Fig. 2** Comparison of experimental functional assays with in silico ncER predictions. **a** CRISPRi effect on cell viability (75,622 sgRNA probes pairs) from Fulco et al.[32] and the corresponding maximum ncER score across the *GATA1* and *MYC* loci. Accuracy at four ncER thresholds is shown in yellow, orange, red and dark-red respectively for the 95th, 99th, 99.5th and 99.9th percentiles. **b** Distribution of maximum ncER at different bins of cell viability (0 to less than −3 log2 fold change). The boxplot's central line represents the median, the bounds represent the 25th and 75th percentile, and the whiskers extend up to 1.5 the interquatile range from the respective bounds. *p* Values were computed with independent two-group Mann–Whitney unpaired test. **c** Fraction of regions with at least one high ncER score, as defined by four different ncER percentile thresholds. CRESTseq peaks are shown in red (N = 45), random matched sized in silico regions extracted from the same locus ("random same locus, N = 4500) in dark gray and random matched sized in silico regions extracted genome-wide ("random GW", N = 4500), in light gray. *p* Values were computed with Fisher's exact test. FC fold change, CRESTseq *cis*-regulatory elements by tiling-deletion and sequencing, GW genome-wide, CRISPRi clustered regularly interspaced short palindromic repeats interference

and generalization using independent sets of human non-coding disease variants.

**Functional correlates of highly ranked non-coding regions**. To assess whether ncER signals effectively identified important non-coding regions, we analyzed two sets of functional data. The first analysis used high-throughput CRISPRi data from 1.29 Mb of sequence in the vicinity of two essential transcription factor genes, *GATA1* and *MYC*[32]. The library deployed more than 80,000 single guide RNA (sgRNAs) pairs tiled across the genomic loci. The readout of the study was cellular proliferation of K562 erythroleukemia cells. To profile the region, we extracted the highest ncER score for each probe. *GATA1* or *MYC* regions were characterized by a high ncER score, with the median being in the 94th percentile (Fig. 2a, b and Supplementary Fig. 8)—which is consistent with the biological importance of the loci. Within this region, we observed a further shift to a median 99th ncER percentile for the regions targeted by the functional pairs of sgRNA probes (*p* < 2.2e−16, compared to non-functional probes—those with less than twofold change decrease in cell viability). Importantly, we removed all sgRNA probes that overlapped with an exon (1746 of 77,368, 2.3%), Fig. 2b. The parameters of predictive performance and accuracy of ncER are shown in Supplementary Table 1. In summary, the sequences that control cellular proliferation, cell viability and gene expression of *GATA1* and *MYC* reside in the regions with high ncER scores.

The second dataset corresponded to high-throughput scanning for *cis*-regulatory elements by tiling-deletion and sequencing (11,570 CREST-seq probes)[33]. The area investigated encompassed 2-Mb of the *POU5F1* locus in human embryonic stem cells. *POU5F1* encodes a transcription factor that plays a key role in embryonic development and stem cell pluripotency. Knockout of *POU5F1* is associated with embryonic mortality in the mouse and scores as an essential gene in humans (pLI score of 0.89)[9,30]. Thus, *POU5F1* is expected to use regulatory elements with features of essentiality[24]. The parameters of predictive performance and accuracy of ncER are shown in Supplementary Table 2 and Supplementary Fig. 9. By extracting peaks of signal, CREST-seq identified 45 *cis*-regulatory elements, including 17 previously annotated as promoters of unrelated genes that, like typical enhancers, form extensive spatial contacts with the *POU5F1* promoter. The CREST-seq peaks encompassed a wide range of sizes and were longer than the probed deletions, therefore increasing the likelihood of having at least one nucleotide with high ncER percentile within the peaks. To account for the longer size, we averaged the ncER signal over 10 bp bins and reextracted the percentiles genome-wide. These binned scores were used for the remainder of the analyses. The 45 enhancers of *POU5F1* were significantly more likely have high ncER scores compared to random genomic loci of matched size (Fig. 2c). For example, 31% of *POU5F1* enhancers reside in regions with ncER >99th percentile, compared to 7–16% for random genomic regions

matched to enhancer size, $p \leq 0.01$. Similarly, the enhancers contained the highest scored regions within the locus in permutation analysis (Supplementary Fig. 10).

In summary, ncER has a good performance for the identification of deleterious variants in the non-coding genome. ncER can also identify non-coding regions associated with cell viability, an in vitro surrogate of essentiality[9], and with regulation of an essential gene. Thus, we speculated that ncER may help map critical regulatory and structural elements of the non-coding genome in the setting of human disease.

**Mapping important regulatory elements in clinical diseases.** We hypothesized that severe genetic diseases that do not have causal variants in the coding region could result from damage to critical non-coding functional elements. To investigate this concept, we chose two different models that represent challenges for the accurate mapping of functional sites in relation to disease: (i) the identification of the critical areas within non-coding structural variants/deletions associated with ASD, and (ii) the impact of reorganization of topologically associated domains (TADs) in the setting of a human developmental disorder that has been modeled in the mouse.

For the first disease model, we assessed a set of cis-regulatory structural variants that were associated with ASD in our previous study (Brandler et al.)[7]. We evaluated deletions from 120 probands and from 16 unaffected siblings. The median ncER percentile were not higher in probands compared to unaffected siblings (median percentile 59.0 compared to 76.1, $p = 0.11$, independent two-group Mann–Whitney unpaired test). However, ASD probands were more likely to carry structural variants with localized high functional domains compared to healthy siblings (Fig. 3a). For example, 25% of the deletions in probands contained regions with ncER > 99th percentile compared to 19% for deletions in unaffected siblings (n.s.), and 11% in 13,600 random genomic regions matched to the size of the deletions ($p = 8.9e{-}06$, Fisher's exact test). The fraction of deletions with high ncER score domain (Fig. 3a) and the cumulative number of nucleotides with high ncER score were consistently higher in ASD proband (Supplementary Figs. 11 and 12). In summary, we observe a general trend of enrichment for high ncER regions in deletions in probands (Supplementary Fig. 13) and propose their location (Fig. 3b). This is consistent with recent data on potentially disruptive mutations among conserved non-coding sequences and a call for improved variant detection and functional classification of non-coding variants[34]. Because the ncER score is trained with severe deleterious and unfitting variants, it may be less sensitive to changes observed in candidate indels in ASD.

Next, we chose a human disease that involves the rearrangement of the regulatory landscape of *IHH* (encoding Indian hedgehog). Brachydactyly A1 is characterized by developmental defects including craniosynostosis and synpolydactyly[35,36]. Will et al.[37] identified nine enhancers with individual tissue specificities in the digit anlagen, growth plates, skull sutures and fingertips. The *IHH* region in humans shares a common structure with the mouse locus that was used for the model by Will et al.[37]. In their study, consecutive deletions resulted in growth defects of the skull and long bones that confirmed that the enhancers function in an additive manner. Deletions and duplications caused dose-dependent upregulation and misexpression of *Ihh*, leading to abnormal phalanges, fusion of sutures and syndactyly. We identified the critical enhancers to reside in an extensive region of high ncER scores, including for the regions shared across human duplications associated with disease (Fig. 3c, blue box). Within the locus, the critical enhancers were also endowed

with high ncER scores (Fig. 3d). For example, 100% of the enhancers were at ncER > 99th percentile, compared to 10% for random genomic regions matched to size, $p = 1.1e{-}09$, Fisher's exact test (Supplementary Fig. 14).

The value of a predictive model that generates scores of pathogenicity depends of the validity of the training sets of pathogenic variants and of the genomic features. The performance of ncER is thus dependent on a limited number of known non-coding pathogenic variants and on the quality of annotation. Our work uses numerous training and validation strategies: independent sets of ClinVar/HGMD and curated Mendelian variants, ASD structural variants, and GWAS variants, as well as functional assessment on two sets of experimental functional data and on a model of craniosynostosis and synpolydactyly. The score can also be limited by ascertainment by proximity: the non-coding pathogenic variants are close to coding regions because that is where they may be primarily searched for. However, our work supports that ncER generalizes to sets of variants with different genomic distributions. We also acknowledge the incomplete orthogonal nature of some of the informative features used in the models. However, Gradient Boosting Trees are well equipped to handle redundant information. There is no doubt that the model will improve in precision with the inclusion of novel features and increasing numbers of annotated non-coding genome variants.

More generally, this work aims at ranking non-coding regions for downstream analysis. We have recently reported on the transcription factor (Myogenic Differentiation 1) MYOD-directed re-configuration of chromatin interactions. We show that MYOD-DNA binding is favored at highly constrained genomic sequences enriched in pathogenic variants[38]. This and recent work[39] contribute to the debate about reconciling redundancy and conservation in the non-coding genome. Work on developmentally expressed genes supports the concept of functionally redundant enhancers in mammalian genomes[40]. Osterwalder et al.[40] indicates that redundancy reduces the likelihood of severe consequences resulting from genetic or environmental challenge. However, Osterwalder et al.[40] also suggest that enhancers can be under purifying selection over evolutionary time and be relevant for organismal fitness under specific pressures because of their contribution to overall gene expression levels. We have in the past indicated that essential genes will use proximal and distant regulatory elements that are conserved and constrained—thus representing putative essentiality in the non-coding genome[9,24]. The current model supports the prioritization of variants and regions across the non-protein coding human genome for diagnostics and for functional analysis.

## Methods

**Training features**. To train the model, we leveraged a total of 38 features from four major categories: (i) gene essentiality, (ii) 3D chromatin structure, (iii) gene expression and other regulatory/functional data and (iv) existing variant pathogenicity/deleteriousness scores. A complete list of features along with their descriptions and accession links can be found in Supplementary Data 2.

We have previously identified a coordination of constrains between genes and their respective *cis* and distal regulatory elements[24]. We implement this concept in the present study by including the following essentiality features: (i) CDTS (our recently developed approach to score the non-coding genome essentiality, based on human genetic diversity[24], (ii) pLI[30], (iii) haploinsufficiency score[41], (iv) gene dosage sensitivity score from ClinGen[41,42], (v) autosomal dominant or recessive categorization[43,44] and (vi) Online Mendelian Inheritance in Man (OMIM) association[45]. For the metrics that solely provide scores for the genic portion of the genome, the respective essentiality features were calculated by mapping each non-coding genomic position to the nearest gene and assigning the corresponding essentiality metric score to the genomic position.

Chromatin 3D structure features included (i) nucleosome positioning extracted from MNase data (https://www.encodeproject.org/), (ii) multiple cell type anchor, loop and domain regions extracted from Hi-C data[46], (iii) frequently interacting regions (FIRE) and TADs extracted from Schmitt et al.[47] and (iv) distal enhancer-TSS

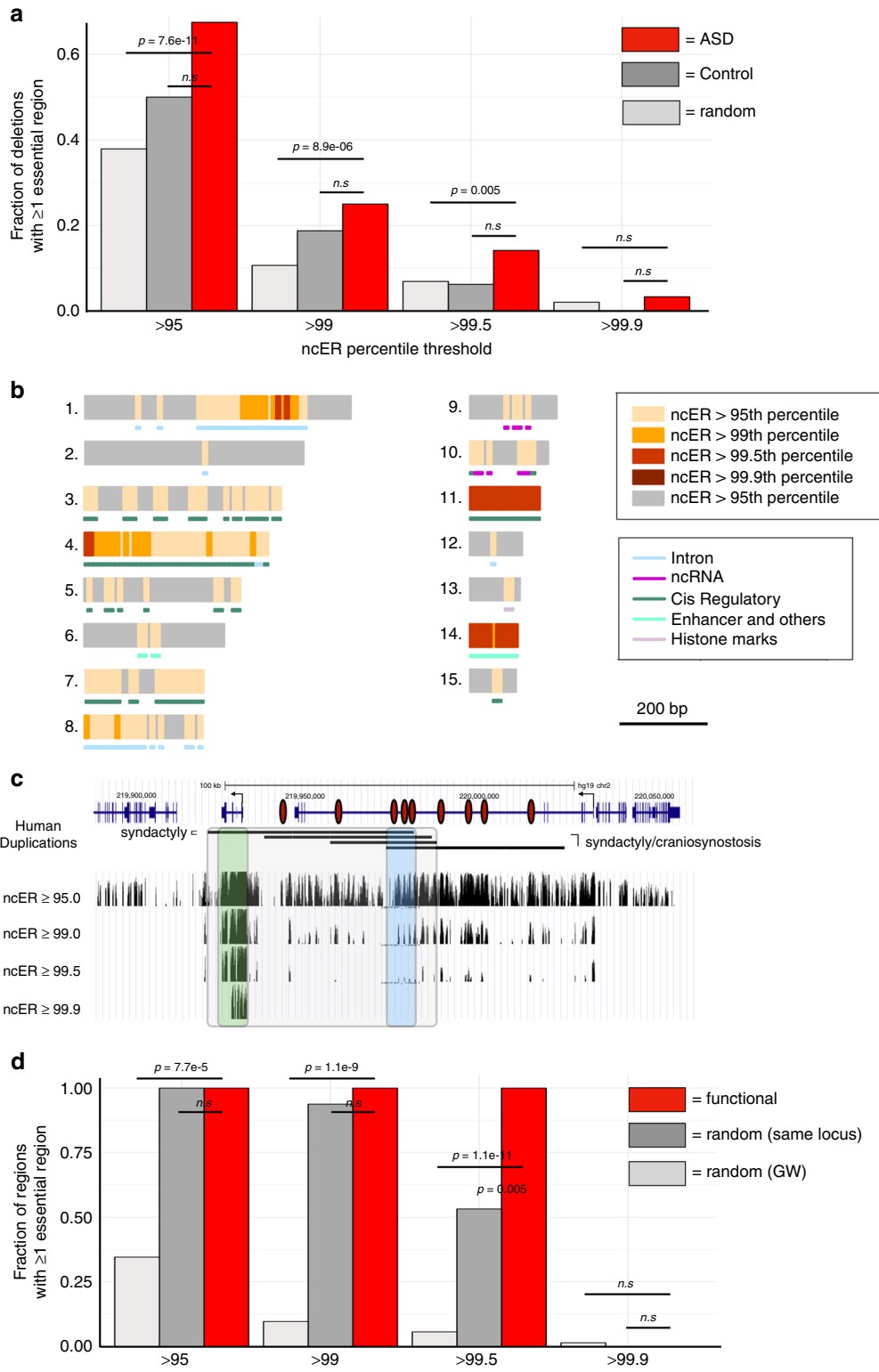

associations extracted from CAGE pairwise expression correlation (FANTOM)[48]. The 3D organization features were either used as binary indicators to denote whether or not a given non-coding genomic position physically interacted with gene promoters, or as discrete values representing the number of cell lines were the structures were identified. Finally, to combine both essentiality and chromatin structure features, we created distal essentiality features, by attributing the respective coding gene essentiality score (pLI) to distal regulatory elements identified through pcHi-C or CAGE pairwise expression correlations.

The model used gene expression, long non-coding RNA (lncRNA) annotations and functional regulatory data that have not been used by other existing metrics.

Those included (i) median gene expression and variance across tissues (GTEx)[49], (ii) functionally tested genomic regions with enhancer activity identified through (ChIP-)STARR-seq experiments[27,50] or validated with transgenic mice[51] and (iii) lncRNAs identified through CAGE and transcriptome analysis[52].

Lastly, variant pathogenicity/deleteriousness scores used in the model included CADD[14], ncEigen[15], FATHMM[18], FunSeq2[17], LINSIGHT[22], ncRVIS[31], Orion[21] and ReMM[19]. We downloaded pre-computed genome-wide scores for each of these metrics (hg19 reference build). In the minority of cases where a per alternative variant score was provided, we used the most "deleterious" value at each position. For the metrics that solely provide scores for the genic portion of the genome, the

**Fig. 3** Mapping of critical domains in disease models. **a** Fraction of deletions with at least one high score ncER bin, defined at four different ncER percentile thresholds. Autism and ASD deletions are shown in red (ASD, $N = 120$), control deletions in dark gray (control, $N = 16$) and random size-matched in silico deletions extracted genome-wide in light gray (random, $N = 13,600$). See Supplementary Fig. 9 for distribution of other sizes of deletions. $p$ Values were computed with Fisher's exact test. **b** Schematic illustration of 15 unique <1 kb deletions (gray bars) identified in autism and ASD probands that harbor high score ncER regions (highlighted in yellow, orange, red and dark red, based on the ncER thresholds). The corresponding genomic elements are displayed under the deletions. Introns are shown in blue, ncRNA in magenta, *cis* regulatory in dark green, enhancers and others regulatory elements in light green and histone marks in light pink (see Methods for categorization of genomic element classes). **c** The upper panel pictogram is adapted from ref. [37] and illustrates the human *IHH* genomic locus associated with developmental defects including craniosynostosis and synpolydactyly[35,36]. It harbors the nine enhancers identified in mice (from Will et al.[37]), represented in dark red ovale shapes. Lower panel, UCSC genome browser view of the region in the locus. The gray box inset highlights the region of high ncER scores across human pathogenic duplications, the blue box highlights the maximal overlap of genomic lesions in humans, and the green box that includes the *IHH* region present in duplications causing syndactyly Leuken type engineered in Will et al.[37]. **d** Fraction of regions with at least one high score ncER, defined by four different ncER percentile thresholds. Mouse to human mapped enhancers are shown in red ("functional", $N = 9$), random size-matched in silico deletions extracted from the same locus ("random same locus", $N = 900$) in dark gray and random size-matched in silico deletions extracted genome-wide ("random GW", $N = 900$) in light gray. $p$ Values were computed with Fisher's exact test. GW genome-wide, ASD autism spectrum disorder

respective features were calculated by mapping each non-coding genomic position to the nearest gene and assigning the corresponding metric score to the genomic position.

**Training variant set**. We used a total of 10,298 SNVs to train the model. The pathogenic dataset comprised non-coding SNVs, located at least 10 bp from the nearest splice site, obtained from HGMD (2016_R1)[29] and ClinVar (July 2016)[28]. The selection criteria for HGMD SNVs were "DM and high" tags, while for ClinVar, SNVs had to be labeled as "Pathogenic" or "Likely Pathogenic", with star 1 or more and no conflicting assertion. HGMD was further filtered out for variants overlapping SNVs annotated as "benign" or "likely benign" in ClinVar (with star 1 or more and no conflicting assertion). Finally, only variants that were not within 500 bp of another pathogenic variants were retained ($N = 782$). The size of 500 bp was fixed to the non-genic feature with the lowest resolution (CDTS), to prevent overweighting of some genomic regions in the model, but still allowing for sub-genic genomic element resolution. In the presence of a cluster of variants, the variant with the smallest start coordinate was selected, followed by the next variant with at least 500 bp distance from the selected variant, etc. The pathogenic variants used to train the model are provided in Supplementary Data 1 (and referred to as "training" in the variant set column). The control genomic training set consisted of 9516 variants chosen from a larger set of variants that were present in the gnomAD dataset at an allelic frequency >1% and matched for the distance to the nearest splice sites and genomic elements. The matching was performed as follows: all pathogenic variants and gnomAD variants with allelic frequency >1% were annotated with their respective distance to the closest splice site and the genomic element they mapped to (see Reference build, annotation and genomic element categorization section). For each pathogenic variant, the subset of control variants falling within the same genomic element was extracted. Within this subset, the closest 15 control variants with the most similar distance to splice site as compared to the pathogenic variant were kept. Finally, duplicated control variants (if any) and falling within 500 bp of another control variant were removed from the final set. A total of 80%/20% of the variants were, respectively, used as training/test sets. In addition, to prevent overfitting, the training/test set variants were split per chromosome regions as follows: the test set encompassed pathogenic variants located on chromosomes 1 (upstream-centromeric region), 10, 19 (upstream-centromeric region), 6 (downstream-centromeric region) and X (downstream-centromeric region), while the training set included the remainder of the chromosomes and/or chromosomic regions. The pathogenic variants used to test the model are provided in Supplementary Data 1 (and referred to as "test" in the variant set column).

**Machine learning model**. We trained an XGBoost model in order to differentiate between pathogenic and control genomic positions in our training set. Hyperparameters were tuned using fivefold cross validation and a randomized search method. A total of 1000 sets of randomly selected hyperparameters were evaluated using fivefold cross validation, and the model that achieved the highest ROC-AUC score was selected. These hyperparameters were then used to train the final model on the entirety of the training set. After hyperparameter tuning, we found that using 32 estimators, a maximum depth of 31, a learning rate of 0.31, and a minimum child weight of 6.17 maximized model performance. We evaluated our model with ROC AUC and PR AUC on the test set (representing 20% of the data).

We annotated each position in the genome with our set of features and used the tuned XGBoost model to make a functionality prediction at each genomic position to score the entire genome. Of note, the model is trained to assess variants and regions of the non-coding genome, and therefore is not relevant for the scoring of protein coding regions.

**Validation sets**. The generalization of the model was assessed on two independent sets of variants. The non-coding pathogenic sets included 209 and 77 new HGMD and ClinVar variants[29] mapping outside/inside ncRNA genes (HGMD 2017_R2 and ClinVar January 2018), at least 500 bp from any pathogenic variants from the training and test sets and from one each other. The control genomic validation sets consisted respectively of 2,090 and 770 variants in the gnomAD dataset at an allelic frequency > 1% and matched to the pathogenic sets for the distance to the nearest splice sites and genomic elements as explained above and at least 500 bp from any control variants from the training and test sets and from one each other. The pathogenic variants used to validate the model are provided in Supplementary Data 1 (and referred to as "generalization_other" and "generalization_ncRNA" in the variant set column).

**Mendelian variants**. To explore ncER percentile distribution of highly likely pathogenic variants, we used a manually curated set of pathogenic non-coding variants associated with Mendelian traits[24], and selected those falling at least 500 bp from any pathogenic variants from the training/test sets, yielding a set of 85 dominant and 52 recessive non-coding pathogenic variants. The pathogenic Mendelian variants are provided in Supplementary Data 3. The control genomic variants ($N = 13,659$) consisted of singleton variants from the gnomAD whole-genome sequencing datasets matched to the pathogenic sets for the distance to the nearest splice sites and genomic elements as explained above.

**GWAS variants**. We used GWAS hit SNVs from GWAS catalog (https://www.ebi.ac.uk/gwas; downloaded on 4 April 2018). We parsed the data to retain only the most significant SNV per locus per phenotype and per study and keep a maximum of one variant per genomic coordinate, yielding to 1785 phenotype-associated variants. GWAS variants are provided in Supplementary Data 1 (and referred to as "gwas" in the variant set column). The control genomic variants ($N > 5$ million) consisted of common variants (allelic frequency > 0.05) from the gnomAD whole-genome sequencing datasets and matched to the phenotype-associated set for the distance to the nearest splice sites and genomic elements as explained above.

**Reference build, annotation and genomic element categorization**. All input features and the model were mapped to the human reference build hg19. To investigate the element distribution, we built an annotation track that combined annotations from GenCode (v.27 mapped to GRCh37) and ENCODE (annotated features and multicell regulatory elements, Ensembl v91 Regulatory Build) and used a prioritization scheme to assign each genomic position a single annotation category (described in ref. [24]). In short, the prioritization was as follows: CDS > Intron (*cis*—within 10 bp of a splice site) > ncRNA > UTR > multicell regulatory > Intron (distal—more than 10 bp of a splice site) > annotated features. For Fig. 3, *Intron* refers to intronic regions (from protein coding or non-coding genes), *ncRNA* refers to exonic regions of non-coding RNAs, *Cis Regulatory* encompasses promoters and untranslated regions (UTRs), *Enhancers and Others* encompasses promoter flanking regions, enhancers, open chromatin, CTCF and other transcription factor binding sites, *Intergenic* refers to unannotated regions and *Histone marks* encompasses H3K9me3 and/or H3K27me3 as well as other histone marks combinations.

**ncER score**. Two sets of ncER percentile scores were computed. The first with nucleotide resolution and the second where raw ncER scores were averaged over 10 bp bins and then expressed as percentiles genome-wide. Both sets of percentiles are provided at https://github.com/TelentiLab/ncER_datasets and can be browsed directly at OMNI (https://www.ai-omni.com/). Intersection of ncER score with other datasets was performed using bedops utility (v2.4.30)[53].

**External datasets**. The CRISPRi coordinates and scores used for analyses displayed in Fig. 2a, b and Supplementary Fig. 8 were obtained from Fulco et al.[32] (http://science.sciencemag.org/highwire/filestream/686019/field_highwire_adjunct_files/2/aag2445_Table_S2.xlsx). As recommended in their paper, the CRISPRi scores were smoothed over 20 subsequent pairs. Only regions that were assessed by at least 20 different pairs and <1 kb long (to prevent size biases) were retained for analysis, resulting in a total of $N = 77,368$ remaining probed pairs. After removing probes overlapping with protein coding region, we retained 75,622 sgRNA pairs.

CREST-seq peaks coordinates used for analyses displayed in Fig. 2c and Supplementary Fig. 10 were obtained from Diao et al.[33] (https://media.nature.com/original/nature-assets/nmeth/journal/v14/n6/extref/nmeth.4264-S7.xlsx). Statistical enrichment of the 11,570 tested sgRNA pairs used for analyses in Supplementary Fig. 9 were also obtained from ref. [33] (https://media.nature.com/original/nature-assets/nmeth/journal/v14/n6/extref/nmeth.4264-S5.xlsx). The locus used for random extraction of same size regions was chr6:30132133–32138339. The matched size random extraction (both in the same locus and genome-wide) was performed 100 times.

*Cis* regulatory transmitted deletions used for analyses displayed in Fig. 3a, b and Supplementary Figs. 11–13 were obtained from Brandler et al.[7] (http://science.sciencemag.org/highwire/filestream/708877/field_highwire_adjunct_files/9/aan2261_TableS7.xlsx, Replication CR Trans sheet). The matched size genome-wide random extraction was performed 100 times.

The mouse enhancer data used for analyses displayed in Fig. 3c and Supplementary Fig. 14 were obtained from Will et al.[37] (https://media.nature.com/original/nature-assets/ng/journal/v49/n10/extref/ng.3939-S1.pdf, Supplementary Table S4). The mouse coordinates were mapped to human using CrossMap (v.0.2.5. http://crossmap.sourceforge.net/) using the mm9ToHg19.over.chain.gz chain. When the mouse enhancers were mapped discontinuously to the human genome, the leftmost and rightmost coordinates in the human genome were used as start and end, respectively. The locus used for random extraction of same size regions was chr2: 219940039–220025587. The matched size random extraction (both in the same locus and genome-wide) was performed 100 times.

**Statistics**. Statistical analyses and plotting were performed with R v3.4.3 (https://www.R-project.org/), notably using the package ggplot2 (http://ggplot2.org/). Data mining was performed using Python (v.2.7.11). The performance predictors in Figs. 1 and 2, Supplementary Figs. 3, 4, 8 and 9, and Supplementary Tables 1 and 2 were assessed as follows: sensitivity or true-positive rate or recall is (TP/(TP + FN)) × 100, specificity is (TN/(TN + FP)) × 100, false-positive rate is (FP/(FP + TN)) × 100, accuracy is ((TP + TN)/(TP + TN + FP + FN)) × 100, positive predictive value or precision is (TP/(TP + FP)) × 100 and negative predictive value is (TN/(TN + FN)) × 100, where TP is a true positive, TN is a true negative, FP is a false positive and FN is a false negative. For the comparison of proportions we used Fisher's exact test; for the comparison of distributions, we used independent two-group Mann–Whitney unpaired test or Kolmogorov–Smirnov test.

**Reporting summary**. Further information on research design is available in the Nature Research Reporting Summary linked to this article.

## Data availability

Genome-wide ncER scores are also provided for download at https://github.com/TelentiLab/ncER_datasets and can be browsed directly at OMNI (https://www.ai-omni.com/).

## Code availability

Code for the model is provided at https://github.com/TelentiLab/ncER_datasets.

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

## Acknowledgements

We thank J. Fellay and A. Rausell for comments on the manuscript. Work of A.Te. is supported by the Qualcomm Foundation and the NIH Center for Translational Science Award (CTSA, grant number UL1TR002550).

## Author contributions

Conception and design of the study: J.d.I, A.Te. Performed the analyses: A.W., J.d.I. Built the browser and code repositories: L.Y. Contributed methods and analytical strategies: D.H., A.To., J.S., B.R. Wrote the manuscript: J.d.I., A.Te.

## Competing interests

The authors declare no competing interests.
