## [Peer Review File · Nature Communications]

Reviewers' Comments:

Reviewer #1:

Remarks to the Author:

Wells and colleagues revised manuscript now under consideration at Nature Communication applies machine learning to model "essential" regulatory DNA sequence in the human genome. This manuscript is an overall strong fit for the broad audience of Nature Communication, as the effort extends previous efforts to computationally address an important question to the fields of disease and evolutionary genetics. There are many limitations regarding this area of modeling, but the manuscript sufficiently addresses these issues. The authors extended their analysis to other datasets, provide genome-wide values of the ncER metric, and developed an online interface for this data. This is an important improvement and makes it much more likely that the results from this work are used in future studies. I suggest addressing two points below. Other than these, I have no major issues and support publication.

Issues:

1. I disagree with the validity of the term "essential" in this context. The algorithm score and quantitative values of the results do not give any information on essentiality in the classical genetics usage. There are many terms that would give the same intended meaning without invoking the specific controlled use of essentiality. Important, critical, etc.

2. For the application of the model across the various datasets (CRISPR, CREST-seq, disease-relevant mutations), it would be of great interest to understand which types of information are driving the ncER score. I brought this up in the initial review, (Rev 1, point 1), and my question may have been interpreted incorrectly. I understand that this might have to be done in a manner parallel to the ncER scoring, e.g. showing the values from the various ncER data input rather than specifically deconstructing the ncER score for given loci. I still think this is very important, as knowing why a genomic interval was identified as critical gives key context for understanding both biology and the value of modeling. For example, it would be great to show in supp or in one of the main figs what input data drove the ncER scores.

Reviewer #2:

Remarks to the Author:

The manuscript "Identification of essential regulatory elements in the human genome" by Wells et al. describes a new computational score trained on known pathogenic vs common variants (>1% AF) for "non-coding" sequences. While multiple methods of a similar study design already exist, the manuscript is of interest due to the application of new feature sets. Further, authors are creative in using various validation sets, however with limited success/persuasive power.

I have read all responses and appreciate the edits to the manuscript. Btw, I am still missing the "control" part of the training data in Suppl. Figure 2.

After giving the manuscript a fresh read after seeing it last about half a year ago, I feel that it still overstates the achievements and that I, as a reader, are not treated with the required honesty. I

will list some points that create this impression below:

- It starts with the title that suggests that I will learn about "identification of essential regulatory elements", however that is not what the manuscript describes. It is about ranking positions in the non-coding part of the genome. Probably less than 1/5 of the non-coding genome constitutes regulatory elements. Further, the method is not about elements but about single positions, more than 50% of the "essential positions" identified by the authors do not have an "essential" neighbor. Further, while essentiality is defined in the manuscript and measures are used that are known to correlate with it, there is no direct evidence that the highly essential positions are actually essential.

- It continues with an abstract highlighting the validation data sets. We hear about CRISPR screens and rearranged topological domains linked to human developmental disorders, so I am excited. In the manuscript, we learn that the CRISPR screens are very targeted screens around three genes and covering at most 1% of the genome. Even more disappointing, we only get signals of enrichment for those. Further, we do not get any comparison to other methods and whether the enrichment is at least higher than for previous methods. When we look in the supplement, we see that there are issues with sensitivity (as predicted before), but nothing discussed about that. Then we continue with the disease applications and despite the ASD results making up more than half a page, the increased burden is not even significant. The only thing significant is a poorly motivated comparison to randomly drawn genomic segments, which is not relevant as we know that SVs are not randomly distributed. The next data set of the IHH locus, is similarly frustrating. Not stated that way, but what we learn is that there are many positions of high predicted "essentiality" in this region and that the experimental identified ones are overlapping it. Still, the concluding sentence turns it around and suggests that ncER would have also highlighted the positions that were experimentally identified – something that was not shown by the authors.

- Another topic is a multiple mention of "redundancy and conservation" or "3D organization" of regulatory sequences. I appreciate that the authors included features that are related to 3D organization and features related to redundancy (promoter/enhancer links) but it remains unclear how the authors are contributing new insights regarding these topics in their study.

- Finally, we end on the claim "We now show hallmarks of proximal or distal regions that regulate the expression of medically important genes. The current model supports the prioritization of variants and regions across the non-protein coding human genome for diagnostics and for functional analysis." and I wonder whether the validation data sets really showed that. Further, later in the methods, I find: "Of note, the model is trained to assess the functionality and essentiality of regulatory regions and should therefore be interpreted as such and therefore is not relevant in protein coding regions." So how exactly do you imagine the steps for it to be used in diagnostics (run coding variant analysis on allele level first, if no candidate is found prioritize variants by ...)?

So I personally think that the manuscript needs a little more honesty about what was achieved and should try to do less overselling. It does not help you if your reader is attracted to your paper by title/abstract, but disappointed after reading the full manuscript.

Reviewers' comments NCOMMS-19-17203-T

Reviewer #1

Wells and colleagues revised manuscript now under consideration at Nature Communication applies machine learning to model “essential” regulatory DNA sequence in the human genome. This manuscript is an overall strong fit for the broad audience of Nature Communication, as the effort extends previous efforts to computationally address an important question to the fields of disease and evolutionary genetics. There are many limitations regarding this area of modeling, but the manuscript sufficiently addresses these issues. The authors extended their analysis to other datasets, provide genome-wide values of the ncER metric, and developed an online interface for this data. This is an important improvement and makes it much more likely that the results from this work are used in future studies. I suggest addressing two points below. Other than these, I have no major issues and support publication.

ANSWER: We thank the reviewer for the supporting comments.

Issues:

Q1.1. I disagree with the validity of the term “essential” in this context. The algorithm score and quantitative values of the results do not give any information on essentiality in the classical genetics usage. There are many terms that would give the same intended meaning without invoking the specific controlled use of essentiality. Important, critical, etc.

ANSWER: We now use “high score ncER” term across the text. We still refer to putative essentiality in the title, and when referring to specific metrics of essential genes. Essentiality is still defined in the text, as it refers to loss of fitness of the organism (deleterious variants). The abstract now states:

TEXT

“The model (ncER, non-coding Essential Regulation) ranks variants in the non-coding genome according to their predicted deleteriousness. ncER can also prioritize non-coding regions associated with regulation of important genes and with cell viability, an in vitro surrogate of essentiality.”

Q1.2. For the application of the model across the various datasets (CRISPR, CREST-seq, disease-relevant mutations), it would be of great interest to understand which types of information are driving the ncER score. I brought this up in the initial review, (Rev 1, point 1), and my question may have been interpreted incorrectly. I understand that this might have to be done in a manner parallel to the ncER scoring, e.g. showing the values from the various ncER data input rather than specifically deconstructing the ncER score for given loci. I still think this is very important, as knowing why a genomic interval was identified as critical gives key context for understanding both biology and the value of

modeling. For example, it would be great to show in supp or in one of the main figs what input data drove the ncER scores.

ANSWER: Figure 1C shows the contributions of the various features to the model. We now follow the advice of the reviewer by illustrating how the various features contribute to the scores across the *IHH* locus (Suppl. Fig. S15).

Reviewer #2

Q2.1. The manuscript "Identification of essential regulatory elements in the human genome" by Wells et al. describes a new computational score trained on known pathogenic vs common variants (>1% AF) for "non-coding" sequences. While multiple methods of a similar study design already exist, the manuscript is of interest due to the application of new feature sets. Further, authors are creative in using various validation sets, however with limited success/persuasive power.

ANSWER: We thank the reviewer for this overview. Although we discuss specific criticisms below, we wish to highlight the following:

(i) the ensemble approach was chosen to capture the power of previous methods. In that sense, this is not one more metric, but the learning from metrics and new features. The comparative performance and the benefit of ensemble approaches is shown in Figure 1A and B. (ii) the manuscript trains, tests and validates on non-coding deleterious variants. Thus, to avoid a wrong message, the term "validation" is only used when presenting the model, while the terms "functional correlates" and "user cases" are applied when exploring the signal of the score across various settings (eg., CRISPR screens, ASD indels). This is consistent with Suppl. Fig. S1, Study design.

Q2.2. I have read all responses and appreciate the edits to the manuscript. Btw, I am still missing the "control" part of the training data in Suppl. Figure 2.

ANSWER: This is now added to Suppl. Fig. S2. As expected, control variants have a very similar distribution because their attributes were closely matched.

Q2.3. After giving the manuscript a fresh read after seeing it last about half a year ago, I feel that it still overstates the achievements and that I, as a reader, are not treated with the required honesty. I will list some points that create this impression below:

ANSWER: we are extremely attentive to this criticism. In our view, the reviewer is questioning our scientific interpretation of the findings, asking for a better recognition of limits, and for stronger support of our claims. Certainly, this is not a question of dishonesty, that we interpret as meaning a purposeful disposition to deceive.

Q2.4. It starts with the title that suggests that I will learn about "identification of essential regulatory elements", however that is not what the manuscript describes. It is about

ranking positions in the non-coding part of the genome. Probably less than 1/5 of the non-coding genome constitutes regulatory elements.

ANSWER: We understand that the reviewer may have a preference for a title that refers to the non-coding genome and not to the regulatory genome, and we have modified the title accordingly. However, we disagree on the comment that “that is not what the manuscript describes”. Suppl. Fig. S5 specifically describes the ranking of the regulatory regions of the genome (as defined by Ensembl). The emphasis on the regulatory genome was on the basis of the high enrichment for this category at high scores compared to other categories (eg. Intronic sequences). The manuscript now generally refers to non-coding genome rather than to regulatory genome.

Q2.5. Further, the method is not about elements but about single positions, more than 50% of the "essential positions" identified by the authors do not have an "essential" neighbor. Further, while essentiality is defined in the manuscript and measures are used that are known to correlate with it, there is no direct evidence that the highly essential positions are actually essential.

ANSWER: We identify single positions and regions/bins of putative essentiality. The definition of essentiality is, as indicated by the reviewer, stated in the first two lines of the abstract and in the text. We now modified the title to indicate “putative essential regions”. We have now limited the use of the term “essential” throughout the text. Regarding whether regions with high score may represent essential genomic elements, we include the following text:

TEXT:

“The model (ncER, non-coding Essential Regulation) ranks variants in the non-coding genome according to their predicted deleteriousness. ncER can also prioritize non-coding regions associated with regulation of important genes and with cell viability, an in vitro surrogate of essentiality.”

“More generally, this work aims at ranking non-coding regions for downstream analysis.”

Q2.6. It continues with an abstract highlighting the validation data sets. We hear about CRISPR screens and rearranged topological domains linked to human developmental disorders, so I am excited. In the manuscript, we learn that the CRISPR screens are very targeted screens around three genes and covering at most 1% of the genome. Even more disappointing, we only get signals of enrichment for those. Further, we do not get any comparison to other methods and whether the enrichment is at least higher than for previous methods. When we look in the supplement, we see that there are issues with sensitivity (as predicted before), but nothing discussed about that. Then we continue with the disease applications and despite the ASD results making up more than half a page, the increased burden is not even significant. The only thing significant is a poorly motivated comparison to randomly drawn genomic segments, which is not relevant as we know that SVs are not randomly distributed. The next data set of the IHH locus, is similarly frustrating. Not stated that way, but what we learn

is that there are many positions of high predicted "essentiality" in this region and that the experimental identified ones are overlapping it. Still, the concluding sentence turns it around and suggests that ncER would have also highlighted the positions that were experimentally identified – something that was not shown by the authors.

ANSWER. As indicated in answer to Q2.1., we now refer to validation to describe the steps after training of the model with non-coding deleterious variants. We then use the scores from the validated model to explore “functional correlates” and we assess applicability with clinical user cases. These are of different nature and allows the reader to think about applications and limits.

Specific to CRISPR and functional screens – we use data that is published and publicly available. To our knowledge there is no published whole genome non-coding CRISPR screen, and we are surprised that the reviewer considers the data insufficient: we use data from 75,822 CRISPR guides covering 1,229,834 bp and 11,570 CRESTseq probes that cover 2,006,207 bp.

Similarly, and as indicated in the previous round of reviews, there is a paucity of non-coding genome data for ASD, and very few studies on human diseases associated with rearrangements of the genome. We have shortened the text describing the ASD analyses. We also indicate that candidate ASD lesions may have characteristics that differentiate them from the unfitting deleterious variants that are used to train the model; thus, explaining the lesser sensitivity (see text below).

We chose *IHH* because of the availability of mouse model data that dissects the various enhancer elements of *IHH*. It is thus surprising to request more validation data where there is, by evidence, little.

Analysis of sensitivity would be helped by having more pathogenic mutations, and more data on functionally tested essential regions. Supp. Fig. S3 and S4 show the performance of the score compared to individual existing metrics. Suppl. Tables S4 and S5 show the predictive performance and accuracy of ncER compared to functional assays.

TEXT:

“Because the ncER score is trained with severe deleterious and unfitting variants, it may be less sensitive to changes observed in candidate indels in ASD.”

Q2.7. Another topic is a multiple mention of "redundancy and conservation" or "3D organization" of regulatory sequences. I appreciate that the authors included features that are related to 3D organization and features related to redundancy (promoter/enhancer links) but it remains unclear how the authors are contributing new insights regarding these topics in their study.

ANSWER: We have now limited the discussion on redundancy and conservation to the last paragraph of the manuscript. We do not discuss this concept any longer in direct connection with 3D structure – although 3D features contribute to the ncER score (Fig 1C). We now cite a collaboration paper in press in *Molecular Cell* that highlights constraints at topological regions in association with transcription binding remodeling of chromatin interactions.

TEXT: "We have recently reported on the transcription factor MYOD-directed re-configuration of chromatin interactions. MYOD-DNA binding is favored at highly constrained genomic sequences enriched in pathogenic variants [ref 38]. This and recent work contribute to the debate about reconciling redundancy and conservation in the non-coding genome."

Q2.8. Finally, we end on the claim "We now show hallmarks of proximal or distal regions that regulate the expression of medically important genes. The current model supports the prioritization of variants and regions across the non-protein coding human genome for diagnostics and for functional analysis." and I wonder whether the validation data sets really showed that. Further, later in the methods, I find: "Of note, the model is trained to assess the functionality and essentiality of regulatory regions and should therefore be interpreted as such and therefore is not relevant in protein coding regions." So how exactly do you imagine the steps for it to be used in diagnostics (run coding variant analysis on allele level first, if no candidate is found prioritize variants by ...)?

ANSWER: We have removed the phrase on "hallmarks" from the discussion. We maintain the comment on prioritization of variants for diagnosis and testing because this is the current use of ranking scores (eg. CADD) in clinical genetics. We rephrased the comment on ncER not being trained for, nor applicable to coding regions. Regarding the optimal use of ncER, this is stated in the manuscript as follows:

TEXT

"We hypothesized that severe genetic diseases that do not have causal variants in the coding region could result from damage to critical non-coding functional elements."

"Of note, the model is trained to assess variants and regions of the non-coding genome, and therefore is not relevant for the scoring of protein coding regions."

Q2.9. So I personally think that the manuscript needs a little more honesty about what was achieved and should try to do less overselling. It does not help you if your reader is attracted to your paper by title/abstract, but disappointed after reading the full manuscript.

ANSWER: We have now modified the title of the manuscript, clarified the difference between validating using non-coding pathogenic variants, and showcased the correlation of the scores with functional screens and its use in two clinical settings.

Reviewers' Comments:

Reviewer #2:

Remarks to the Author:

This is the third round of review of the manuscript by Alex Wells et al. As a reviewer, I feel we have hit a point of diminishing returns and that we need an editorial decision now. The authors have adjusted their manuscript with respect to the comments of the two reviewers. While I acknowledge the edits, the authors decide not following the recommendations of the reviewers completely. For example, the term "essential" is still used and comparisons to other tools (e.g. for the CRISPR screen enrichment) are still not available for all datasets. However, we have achieved a manuscript that I have no additional comments for and which I can recommend for publication.